# Which Algorithmic Choices Matter at Which Batch Sizes? Insights From a Noisy Quadratic Model

**Guodong Zhang**[1,2,3*], **Lala Li**[3], **Zachary Nado**[3], **James Martens**[4],
**Sushant Sachdeva**[1], **George E. Dahl**[3], **Christopher J. Shallue**[3], **Roger Grosse**[1,2]
[1]University of Toronto, [2]Vector Institute, [3]Google Research, Brain Team, [4]DeepMind

## Abstract

Increasing the batch size is a popular way to speed up neural network training, but beyond some critical batch size, larger batch sizes yield diminishing returns. In this work, we study how the critical batch size changes based on properties of the optimization algorithm, including acceleration, preconditioning and averaging, through two different lenses: large scale experiments, and analysis of a simple noisy quadratic model (NQM). We experimentally demonstrate that optimization algorithms that employ preconditioning, specifically Adam and K-FAC, result in much larger critical batch sizes than stochastic gradient descent with momentum. We also demonstrate that the NQM captures many of the essential features of real neural network training, despite being drastically simpler to work with. The NQM predicts our results with preconditioned optimizers and exponential moving average, previous results with accelerated gradient descent, and other results around optimal learning rates and large batch training, making it a useful tool to generate testable predictions about neural network optimization.

## 1   Introduction

Increasing the batch size is one of the most appealing ways to accelerate neural network training on data parallel hardware. Larger batch sizes yield better gradient estimates and, up to a point, reduce the number of steps required for training, which reduces the training time. The importance of understanding the benefits of modern parallel hardware has motivated a lot of recent work on training neural networks with larger batch sizes [Goyal et al., 2017, Osawa et al., 2018, McCandlish et al., 2018, Shallue et al., 2018]. To date, the most comprehensive empirical study of the effects of batch size on neural network training is Shallue et al. [2018], who confirmed that increasing the batch size initially achieves perfect scaling (i.e. doubling the batch size halves the number of steps needed) up to a problem-dependent critical batch size, beyond which it yields diminishing returns [Balles et al., 2017, Goyal et al., 2017, Jastrzębski et al., 2018, McCandlish et al., 2018]. Shallue et al. [2018] also provided experimental evidence that the critical batch size depends on the optimization algorithm, the network architecture, and the data set. However, their experiments only covered plain SGD, SGD with (heavy-ball) momentum, and SGD with Nesterov momentum, leaving open the enticing possibility that other optimizers might extend perfect scaling to even larger batch sizes.

Empirical scaling curves like those in Shallue et al. [2018] are essential for understanding the effects of batch size, but generating such curves, even for a single optimizer on a single task, can be very expensive. On the other hand, existing theoretical analyses that attempt to analytically derive critical batch sizes (e.g. Ma et al. [2018], Yin et al. [2018], Jain et al. [2018]) do not answer our questions about which optimizers scale the best with batch size. They tend to make strong assumptions, produce parameter-dependent results that are difficult to apply, or are restricted to plain SGD. It would be

---

[*]Work done as part of the Google Student Researcher Program. Email: `gdzhang@cs.toronto.edu`

ideal to find a middle ground between a purely empirical investigation and theoretical analysis by building a model of neural network optimization problems that captures the essential behavior we see in real neural networks, while still being easy to understand. Additionally, we need to study optimizers beyond momentum SGD since they might provide us an approach to exploit speedups from the very largest batch sizes. In this work, we make the following contributions:

1. We show that a simple noisy quadratic model (NQM) is remarkably consistent with the batch size effects observed in real neural networks, while allowing us to run experiments in seconds, making it a great tool to generate testable predictions about neural network optimization.

2. We show that the NQM successfully predicts that momentum should speed up training relative to plain SGD at larger batch sizes, but have no benefit at small batch sizes.

3. Through large scale experiments with Adam [Kingma and Ba, 2014] and K-FAC [Martens and Grosse, 2015], we confirm that, as predicted by the NQM, preconditioning extends perfect batch size scaling to larger batch sizes than are possible with momentum SGD alone. Furthermore, unlike momentum, preconditioning can help at small batch sizes as well.

4. Lastly, we show that, as predicted by the NQM, exponential moving averages reduce the number of steps required for a specific batch size and can achieve the same acceleration with smaller batch sizes, thereby saving computation.

## 2 Related Work

In a classic paper, Bottou and Bousquet [2008] studied the asymptotics of stochastic optimization algorithms and found SGD to be competitive with fancier approaches. They showed that stochastic optimization involves fundamentally different tradeoffs from full-batch optimization. More recently, several studies have investigated the relationship between batch size and training time for neural networks. Chen et al. [2018] studied the effect of network width on the critical batch size, and showed experimentally that it depends on both the data set and network architecture. Golmant et al. [2018] studied how various heuristics for adjusting the learning rate as a function of batch size affect the relationship between batch size and training time. Shallue et al. [2018] conducted a comprehensive empirical study on the relationship between batch size and training time with different neural network architectures and data sets using plain SGD, heavy-ball momentum, and Nesterov momentum. Finally, McCandlish et al. [2018] used the average gradient noise over training to predict the critical batch size. All of these studies described a basic relationship between batch size and training steps to a fixed error goal, which is comprised of three regions: perfect scaling initially, then diminishing returns, and finally no benefit for all batch sizes greater than the critical batch size.

Other studies have attempted to characterize the critical batch size analytically in stochastic optimization. Under varying assumptions, Ma et al. [2018], Yin et al. [2018], Jain et al. [2018] all derived analytical notions of critical batch size, but to our knowledge, all for SGD.

Additionally, previous studies have shown that SGD and momentum SGD are equivalent for small learning rates (after appropriate rescaling), both for the continuous limit [Leen and Orr, 1994] and discrete settings Yuan et al. [2016]. However, they do not explain why momentum SGD (including heavy-ball and Nesterov momentum) sometimes outperforms plain SGD in mini-batch training (as observed by Kidambi et al. [2018] and Shallue et al. [2018]). Concurrently, Smith et al. [2019] showed that momentum outperforms plain SGD at large batch sizes.

Finally, there are a few works studying average of the iterates, rather than working with the last iterate. This is a classical idea in optimization, where it is known to provide improved convergence [Polyak and Juditsky, 1992, Bach and Moulines, 2013, Dieuleveut and Bach, 2016]. However, most of them focused on *tail averaging*, which you have to decide ahead of time the iteration to start accumulating the running averaging. More commonly (especially in deep learning), exponential moving average [Martens, 2014] is preferred for its simplicity and ability to handle non-convex landscape. However, no analysis was done especially when mini-batch is used.

## 3 Analysis of the Noisy Quadratic Model (NQM)

In this section, we work with a *noisy quadratic model* (NQM), a stochastic optimization problem whose dynamics can be simulated analytically, in order to reason about various phenomena en-

countered in training neural networks. In this highly simplified model, we first assume the loss function being optimized is a convex quadratic, with noisy observations of the gradient. For analytic tractability, we further assume the noise covariance is codiagonalizable with the Hessian. Because we are not interested in modeling overfitting effects, we focus on the online training setting, where the observations are drawn i.i.d. in every training iteration. Under these assumptions, we derive an analytic expression for the risk after any number of steps of SGD with a fixed step size, as well as a dynamic programming method to compute the risk following a given step size schedule.

Convex quadratics may appear an odd model for a complicated nonconvex optimization landscape. However, one obtains a convex quadratic objective by linearizing the network's function around a given weight vector and taking the second-order Taylor approximation to the loss function (assuming it is smooth and convex). Indeed, recent theoretical works [Jacot et al., 2018, Du et al., 2019, Zhang et al., 2019a] show that for wide enough networks, the weights stay close enough to the initialization for the linearized approximation to remain accurate. Empirically, linearized approximations closely match a variety of training phenomena for large but realistic networks [Lee et al., 2019].

## 3.1 Problem Setup

We now introduce the noisy quadratic model [Schaul et al., 2013, Martens, 2014, Wu et al., 2018], where the true function being optimized is a convex quadratic. Because we analyze rotation-invariant and translation-invariant optimizers such as SGD and heavy-ball momentum, we assume without loss of generality that the quadratic form is diagonal, and that the optimum is at the origin. Hence, our exact cost function decomposes as a sum of scalar quadratic functions for each coordinate:

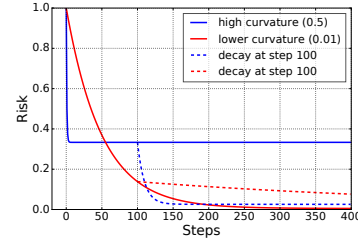

**Figure 1:** Cartoon of the evolution of risk for different coordinates with and without learning rate decay.

$$\mathcal{L}(\boldsymbol{\theta}) = \frac{1}{2}\boldsymbol{\theta}^\top \mathbf{H}\boldsymbol{\theta} = \frac{1}{2}\sum_{i=1}^{d} h_i \theta_i^2 \triangleq \sum_{i=1}^{d} \ell(\theta_i). \qquad (1)$$

Without loss of generality, we assume $h_1 \geq h_2 \geq ... \geq h_d$. We consider a single gradient query to have the form $g(\boldsymbol{\theta}) = \mathbf{H}\boldsymbol{\theta} + \boldsymbol{\epsilon}$ where $\mathbb{E}[\boldsymbol{\epsilon}] = \mathbf{0}$ and $\mathrm{Cov}(\boldsymbol{\epsilon}) = \mathbf{C}$. To reduce the variance of gradient estimation, we can average over multiple independent queries, which corresponds to "mini-batch training" in neural network optimization. We denote the averaged gradient as $g_B(\boldsymbol{\theta})$ and the covariance $\mathrm{Cov}(g_B(\boldsymbol{\theta})) = \mathbf{C}/B$, where $B$ is the number of queries (mini-batch size).

For analytical tractability, we make the nontrivial assumption that $\mathbf{H}$ and $\mathbf{C}$ are codiagonalizable. (Since $\mathbf{H}$ is diagonal, this implies that $\mathbf{C} = \mathrm{diag}(c_1, \dots, c_d)$.) See Section 3.5 for justification of this assumption. Under gradient descent with fixed step size $\alpha$, each dimension evolves independently as

$$\theta_i(t+1) = (1 - \alpha h_i)\theta_i(t) + \alpha\sqrt{c_i/B}\epsilon_i, \qquad (2)$$

where $\alpha$ is the learning rate and $\epsilon_i$ is zero-mean unit variance iid noise. By treating $\theta_i$ as a random variable, we immediately obtain the dynamics of its mean and variance.

$$\mathbb{E}\left[\theta_i(t+1)\right] = (1-\alpha h_i)\mathbb{E}\left[\theta_i(t)\right], \quad \mathbb{V}\left[\theta_i(t+1)\right] = (1-\alpha h_i)^2 \mathbb{V}\left[\theta_i(t)\right] + \frac{\alpha^2 c_i}{B}. \qquad (3)$$

Based on eqn. (3), the expected risk after $t$ steps in a given dimension $i$ is

$$\mathbb{E}\left[\ell(\theta_i(t))\right] = \underbrace{(1-\alpha h_i)^{2t}}_{\text{convergence rate}} \mathbb{E}\left[\ell(\theta_i(0))\right] + \left(1 - (1-\alpha h_i)^{2t}\right)\underbrace{\frac{\alpha c_i}{2B(2-\alpha h_i)}}_{\text{steady state risk}}, \qquad (4)$$

where we have assumed that $\alpha h_i \leq 2$. (Note that this can be seen as a special case of the convergence result derived for convex quadratics in Martens [2014].)

Remarkably, each dimension converges exponentially to a steady state risk. Unfortunately, there is a trade-off in the sense that higher learning rates (up to $1/h_i$) give faster convergence to the steady state risk, but also produce higher values of the steady-state risk. The steady state risk also decreases proportionally to increases in batch size; this is important to note because in the following subsections, we will show that traditional acceleration techniques (e.g., momentum and preconditioning) help improve the convergence rate at the expense of increasing the steady state risk. Therefore, the NQM implies that momentum and preconditioning would benefit more from large-batch training compared to plain SGD, as shown in later sections.

## 3.2 Momentum Accelerates Training at Large Batch Sizes

Applied to the same noisy quadratic model as before, the update equations for momentum SGD are:

$$m_i(t+1) = \beta m_i(t) + h_i \theta_i(t) + \sqrt{c_i/B}\epsilon_i,$$
$$\theta_i(t+1) = \theta_i(t) - \alpha m_i(t+1). \tag{5}$$

We show in the following theorem (see Appendix C for proof) that momentum SGD performs similarly to plain SGD in the regime of small batch sizes but helps in the large-batch regime, which can be viewed as a near-deterministic optimization problem.

**Theorem 1.** *Given a dimension index $i$, and $0 \le \beta < 1$ with $\beta \ne (1 - \sqrt{\alpha h_i})^2$, the expected risk at time $t$ associated with that dimension satisfies the upper bound*

$$\mathbb{E}\left[\ell(\theta_i(t))\right] \le \left(\frac{(r_1^{t+1} - r_2^{t+1}) - \beta(r_1^t - r_2^t))}{r_1 - r_2}\right)^2 \mathbb{E}\left[\ell(\theta_i(0))\right] + \frac{(1+\beta)\alpha c_i}{2B(2\beta + 2 - \alpha h_i)(1-\beta)}, \tag{6}$$

*where $r_1$ and $r_2$ (with $r_1 \ge r_2$) are the two roots of the quadratic equation $x^2 - (1 - \alpha h_i + \beta)x + \beta = 0$.*

As with plain SGD (c.f. eqn. (4)), the loss associated with each dimension can be expressed as the sum of two terms, where the first one decays exponentially and corresponds to the behavior of the deterministic version of the algorithm, and the second remains constant.

Following the existing treatment of the deterministic version of the algorithm [Chiang, 1974, Qian, 1999, Yang et al., 2018, Goh, 2017], we divide our analysis two cases: *overdamping* and *underdamping*. In the case of overdamping, where $\beta < (1 - \sqrt{\alpha h_i})^2$, both roots $r_1$ and $r_2$ are real and therefore the convergence rate is determined by the larger one (i.e. $r_1$), which has the value

$$r_1 = \frac{1 - \alpha h_i + \beta + \sqrt{(1-\beta)^2 - 2(1+\beta)\alpha h_i + \alpha^2 h_i^2}}{2} \tag{7}$$

With a fixed learning rate, the steady state risk will be constant, and the best achievable expected risk will be lower bounded by it. Thus, to achieve a certain target loss we must either drive the learning rate down, or the batch size up. Assuming a small batch size and a low target risk, we are forced to pick a small learning rate, in which case one can show[2] that $r_1 \approx 1 - \alpha h/1 - \beta$. In Figure 2 we plot the convergence rate as a function of $\beta$, and we indeed observe that the convergence rate closely matches $1 - \alpha h/1 - \beta$, assuming a relative small learning rate. We further note that the convergence rate and steady state risk of eqn. (6) are the same as the ones in plain SGD (eqn. (4)), except that they use an "effective learning rate" of $\alpha/1-\beta$. To help validate these predictions, in Appendix E.3 we provide a comparison of momentum SGD with plain SGD using the effective learning rate.

In the case of underdamping where $\beta > (1 - \sqrt{\alpha h_i})^2$, both $r_1$ and $r_2$ will be complex and have norm $\sqrt{\beta}$. We note that the optimal $\beta$ should be equal to or smaller than $(1 - \sqrt{\alpha h_d})^2$, since otherwise all dimensions are underdamped, and we can easily improve the convergence rate and steady state risk by reducing $\beta$.

Next we observe that the convergence of the total loss will *eventually* be dominated by the slowest converging dimension (which corresponds to the smallest curvature $h_d$), and this will be in the overdamping regime as argued above. By our analysis of the overdamping case, we can achieve the same convergence rate for this dimension by simply replacing the learning rate $\alpha$ in the bound for plain SGD (eqn. (4)) with the effective learning rate $\alpha/1-\beta$.

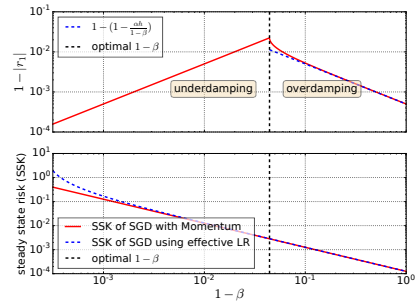

**Figure 2:** Convergence rate and steady state risk (SSK) as a function of momentum for a single dimension with $\alpha h = 0.0005$ and batch size $B = 1$.

So while momentum gives no long-term training acceleration for very low fixed learning rates (which we are forced to use when the batch size is small), we note that it can help in large-batch training. With $\beta > 0$, the steady state risk roughly amplifies by a factor of $1/1-\beta$, and we note that steady state risk also decreases proportionally to increases in batch size. Therefore, we expect momentum SGD to exhibit perfect scaling up to larger batch sizes than plain SGD.

## 3.3 Preconditioning Further Extends Perfect Scaling to Larger Batch Sizes

Many optimizers, such as Adam and K-FAC, can be viewed as preconditioned gradient descent methods. In each update, the gradient is rescaled by a PSD matrix $\mathbf{P}^{-1}$, called the preconditioner.

$$\boldsymbol{\theta}(t+1) = \boldsymbol{\theta}(t) - \alpha\mathbf{P}^{-1}\left[\mathbf{H}\boldsymbol{\theta} + \boldsymbol{\epsilon}\right]. \tag{8}$$

In lieu of trying to construct noisy quadratic analogues of particular optimizers, we analyze preconditioners of the form $\mathbf{P} = \mathbf{H}^p$ with $0 \leq p \leq 1$. Note that $\mathbf{P}$ remains fixed throughout training since the Hessian $\mathbf{H}$ is constant in the NQM. We can recover standard SGD by setting $p = 0$.

Conveniently, for our NQM, the dynamics of preconditioned SGD are equivalent to the SGD dynamics in an NQM with Hessian $\tilde{\mathbf{H}} = \mathbf{P}^{-1/2}\mathbf{H}\mathbf{P}^{-1/2}$ and gradient covariance $\tilde{\mathbf{C}} = \mathbf{P}^{-1/2}\mathbf{C}\mathbf{P}^{-1/2}$. Hence, the dynamics can be simulated using eqn. (4), exactly like the non-preconditioned case. We immediately obtain the following bound on the risk:

$$\mathbb{E}\left[\mathcal{L}(\boldsymbol{\theta}(t))\right] \leq \sum_{i=1}^{d}(1 - \alpha h_i^{(1-p)})^{2t}\mathbb{E}\left[\ell(\theta_i(0))\right] + \sum_{i=1}^{d}\frac{\alpha c_i h_i^{-p}}{2B(2 - \alpha h_i^{1-p})}. \tag{9}$$

To qualitatively understand the effect of preconditioning, first consider the first term in eqn. (8). The convergence of this term resembles that of gradient descent on a deterministic quadratic, which (with optimal $\alpha \approx 2/\tilde{h}_1$) converges exponentially at a rate of approximately $2/\tilde{\kappa}$, where $\tilde{\kappa} = \tilde{h}_1/\tilde{h}_d$ is the condition number of the transformed problem. Since $\tilde{\kappa} = \kappa^{1-p}$, this implies a factor of $\kappa^p$ improvement in the rate of convergence. Hence, for near-deterministic objectives where the first term dominates, values of $p$ closer to 1 correspond to better preconditioners, and result in much faster convergence. Unfortunately, there is no free lunch, as larger values of $p$ will also increase the second term (steady state risk). Assuming an ill-conditioned loss surface ($\kappa \gg 1$), the steady state risk of each dimension becomes

$$\frac{1}{2B}\frac{\alpha c_i h_i^{-p}}{2 - \alpha h_i^{(1-p)}} \approx \frac{c_i}{2Bh_1}\frac{(h_i/h_1)^{-p}}{1 - (h_i/h_1)^{(1-p)}}, \tag{10}$$

which is a monotonically increasing function with respect to $p$. Even without this amplification effect, the steady state risk will eventually become the limiting factor in the minimization of the expected risk. One way to reduce the steady state risk, apart from using Polyak averaging [Polyak and Juditsky, 1992] or decreasing the learning rate (which will harm the rate of convergence), is to increase the batch size. This suggests that the benefits of using stronger preconditioners will be more clearly observed for larger batch sizes, which is an an effect that we empirically demonstrate in later sections.

## 3.4 Exponential Moving Average Reduces Steady State Risk

Following the same procedure as previous two sections, we analyze exponential moving averages (EMA) on our NQM. The update rule of EMA can be written as

$$\begin{aligned}\boldsymbol{\theta}(t+1) &= \boldsymbol{\theta}(t) - \alpha\left[\mathbf{H}\boldsymbol{\theta} + \boldsymbol{\epsilon}\right], \\ \tilde{\boldsymbol{\theta}}(t+1) &= \gamma\tilde{\boldsymbol{\theta}}(t) + (1-\gamma)\boldsymbol{\theta}(t+1).\end{aligned} \tag{11}$$

The averaged iterate $\tilde{\boldsymbol{\theta}}$ is used at test time. The computational overhead is minimal (storing an additional copy of the parameters, plus some cheap arithmetic operations). We now show that EMA outperforms plain SGD by reducing the steady state risk term.

**Theorem 2.** *Given a dimension index $i$, and $0 \leq \gamma < 1$, the expected risk at time t associated with that dimension satisfies the upper bound*

$$\begin{aligned}\mathbb{E}\left[\ell(\tilde{\theta}_i(t))\right] &\leq \left(\frac{(r_1^{t+1} - r_2^{t+1}) - \gamma(1 - \alpha h_i)(r_1^t - r_2^t))}{r_1 - r_2}\right)^2\mathbb{E}\left[\ell(\theta_i(0))\right] \\ &+ \frac{\alpha c_i}{2B(2 - \alpha h_i)}\frac{(1-\gamma)(1 + (1 - \alpha h_i)\gamma)}{(1+\gamma)(1 - (1 - \alpha h_i)\gamma)},\end{aligned} \tag{12}$$

*where $r_1 = 1 - \alpha h_i$ and $r_2 = \gamma$.*

By properly choosing an averaging coefficient $\gamma < 1 - \alpha h_d$ such that $r_1 > r_2$, one can show that EMA reduces the steady state risk without sacrificing the convergence rate. To see this, we note that the red part of eqn. (12) is strictly less than 1 given the fact $1 - \alpha h_i < 1$ while the other part is exactly the same as the steady state risk of plain SGD.

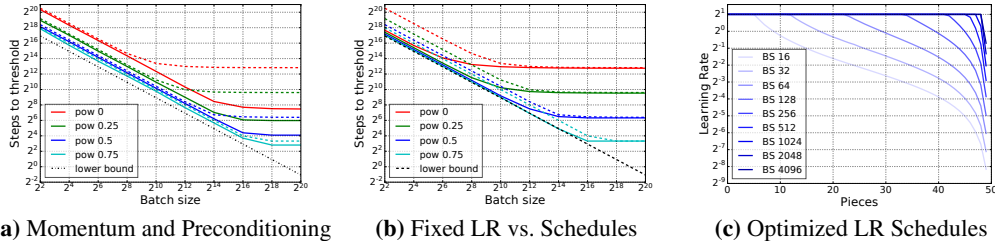

**Figure 3:** (a) **Effects of momentum and preconditioning.** Steps required to reach target loss as a function of batch size under different preconditioning power. Solid lines are momentum SGD while dashed lines are plain SGD. The black dashed line is the information theoretic lower bound. (b) **Effect of learning rate decay.** The solid lines use the optimized piecewise constant scheme, which are shown in (c) for power 0. The dashed curves in (b) are plain SGD for comparison. We observe that learning rate schedules close most of the gap between the fixed learning rate performance and the information theoretic lower bound.

## 3.5 Choice of H and C

We've found that the qualitative behavior of optimizers in our NQM depends on the choices of $\mathbf{H}$ and $\mathbf{C}$. Therefore, we choose matrices motivated by theoretical and empirical considerations about neural net training. First, we set the diagonal entries of $\mathbf{H}$ to be $\{\frac{1}{i}\}_{i=1}^{d}$ for some integer $d$, giving a condition number of $d$. This closely matches the estimated eigenspectrum of the Hessian of a convolutional network (see Figure 9 and Appendix E.4), and is also consistent with recent work finding heavy tailed eigenspectra of neural network Hessians [Ubaru et al., 2017, Ghorbani et al., 2019]. We choose $d = 10^4$, which approximately matches the condition number of the K-FAC Hessian approximation for ResNet8. (Qualitative behaviors were consistent for a wide range of $d$.)

We also set $\mathbf{C} = \mathbf{H}$ (a nontrivial assumption). This was motivated by theoretical arguments that, under the assumption that the implicit conditional distribution over the network's output is close to the conditional distribution of targets from the training distribution, the Hessian closely matches the gradient covariance in neural network training [Martens, 2014]. Empirically, this relationship appears to hold tightly for a convolutional network and moderately well for a transformer (see Appendix E.2).

## 3.6 Information Theoretic Lower Bound

Since our NQM assumes the infinite data (online optimization) setting, it's instructive to compare the performance of optimizers against an information theoretic lower bound. Specifically, under the assumption that $\mathbf{H} = \mathbf{C}$, the NQM is equivalent to maximum likelihood estimation of the mean vector for a multivariate Gaussian distribution with covariance $\mathbf{H}^{-1}$. Hence, the risk obtained by any optimizer can be bounded below by the risk of the maximum likelihood estimator for the Gaussian, which is $d/2N$, where $d$ is the dimension and $N$ is the total number of training examples visited. We indicate this bound with a dashed black line in our plots.

## 3.7 Noisy Quadratic Experiments

In this section, we simulate noisy quadratic optimization using the closed-form dynamics. Our aim is to formulate hypotheses for how different optimizers would behave for neural network optimization. Our main metric is the number of steps required to achieve a target risk. For efficiency, rather than explicitly representing all the eigenvalues of $\mathbf{H}$, we quantize them into 100 bins and count the number of eigenvalues in each bin. Unless otherwise specified, we initialize $\boldsymbol{\theta}$ as $\mathcal{N}(\mathbf{0}, \mathbf{I})$ and use a target risk of 0.01. (The results don't seem to be sensitive to either the initial variance or the target risk; some results with varying target risk thresholds are shown in Appendix E.5).

### 3.7.1 Effect of Momentum, Preconditioning and Exponential Moving Average

We first experiment with momentum and varying preconditioner powers on our NQM. We treat both the (fixed) learning rate $\alpha$ and momentum decay parameter $\beta$ as hyperparameters, which we tune using a fine-grained grid search.

Consistent with the empirical results of Shallue et al. [2018], each optimizer shows two distinct regimes: a small-batch (stochastic) regime with perfect linear scaling, and a large-batch (deterministic)

| Data Set | Size | Model | Remarks | LR |
|---|---|---|---|---|
| MNIST | 55,000 | Simple CNN | Same as Shallue et al. [2018] except without dropout regularization. | Constant |
| FMNIST | 55,000 | | | |
| CIFAR10 | 45,000 | ResNet8 without BN | Same as Shallue et al. [2018]. | Constant |
| | | ResNet32 with BN | Ghost batch norm is used. | Linear Decay |
| | | VGG11 with BN | Ghost batch norm is used. | Linear Decay |
| LM1B | ~30M | Two-layer Transformer | Shallow model in Shallue et al. [2018] | Constant |

**Table 1: Data sets and models used in our experiments.** See Appendix F.2 for full details.

regime insensitive to batch size. We call the phase transition between these regimes the *critical batch size*. Consistent with the analysis of Section 3.2 and the observations of Smith et al. [2018], Shallue et al. [2018], Kidambi et al. [2018], the performance of momentum-based optimizers matches that of the plain SGD methods in the small-batch regime, but momentum increases the critical batch size and gives substantial speedups in the large batch regime. Preconditioning also increases the critical batch size and gives substantial speedups in the large batch regime, but interestingly, also improves performance by a small constant factor even for very small batches. Combining momentum with preconditioning extends both of these trends.

We next experiment with EMA and varying preconditioning powers on our NQM. Following the same procedure as before, we tune both learning rate $\alpha$ and averaging coefficient $\gamma$ using grid search. As expected, EMA reduces the number of steps required especially for plain SGD with preconditioning power 0. Another interesting observation is that EMA becomes redundant in the large batch (near-deterministic) regime since the main effect of EMA is reducing the steady-state risk, which can also be done by increasing the batch size. This implies that EMA would reduce the critical batch size and therefore achieve the same amount of acceleration with less computation.

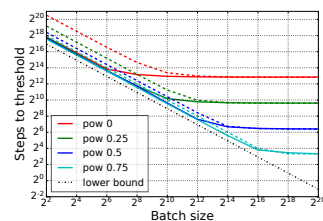

**Figure 4:** Effects of exponential moving average (EMA). Solid lines are SGD with EMA while dashed lines are plain SGD.

### 3.7.2 Optimal Learning Rate and Decay Scheme

In the NQM, we can calculate the optimal constant learning rate given a specific batch size. Figure 14 shows the optimal learning rate as a function of batch size for a target risk of $0.01$. Notably, the optimal learning rate of plain (preconditioned) SGD (Figure 14a) scales linearly with batch size before it hits the critical batch size, matching the scheme used in Goyal et al. [2017]. The linear scaling also holds for the effective learning rate of momentum SGD. In the small batch regime, the optimal effective learning rate for momentum SGD matches the optimal plain SGD learning rate, suggesting that the momentum and learning rate are interchangeable in the small batch regime.

While a fixed learning rate often works well for simple problems, good performance on the ImageNet benchmark [Russakovsky et al., 2015] requires a carefully tuned schedule. Here we explicitly optimize a piecewise constant learning rate schedule for SGD (with 50 pieces), in terms of the number of steps to reach the loss threshold.[3] In Figure 3b, we show that optimized learning rate schedules help significantly in the small batch regime, consistent with the analysis in Wu et al. [2018]. We observe the same linear scaling as with fixed-learning-rate SGD, but with a better constant factor. In fact, optimized schedules nearly achieve the information theoretic optimum. However, learning rate schedules do not improve at all over fixed learning rates in the large batch regime. Figure 3c shows optimized schedules for different batch sizes; interestingly, they maintain a large learning rate throughout training followed by a roughly exponential decay, consistent with commonly used neural network training schedules. Additionally, even though the different batch sizes start with the same learning rate, their final learning rates at the end of training scale linearly with batch size (see Figure 15 in Appendix E.7).

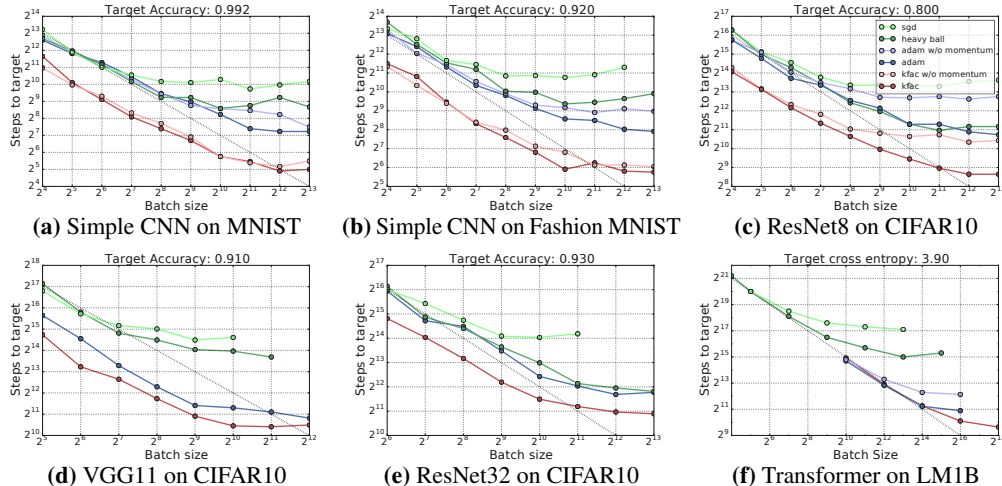

**(a)** Simple CNN on MNIST     **(b)** Simple CNN on Fashion MNIST     **(c)** ResNet8 on CIFAR10

**(d)** VGG11 on CIFAR10     **(e)** ResNet32 on CIFAR10     **(f)** Transformer on LM1B

**Figure 5: Empirical relationship between batch size and steps to result.** Key observations: 1) momentum SGD has no benefit over plain SGD at small batch sizes, but extends the perfect scaling to larger batch sizes; 2) preconditioning also extends perfect scaling to larger batch sizes, i.e. K-FAC > Adam > momentum SGD. This is most noticeable in the Transformer model; 3) preconditioning (particularly K-FAC) reduces the number of steps needed to reach the target even for small batch sizes. All of these agree with the predictions by NQM.

## 4 Neural Network Experiments

We investigated whether the predictions made by the NQM hold in practice by running experiments with five neural network architectures across three image classification tasks and one language modeling task (see Table 1). For each model and task, we compared a range of optimizers: SGD, momentum SGD, Adam (with and without momentum), and K-FAC (with and without momentum). For K-FAC, preconditioning is applied before momentum. See Appendix F for more details.

The primary quantity we measured is the number of steps required to reach a target accuracy (for image classification tasks) or cross entropy (for language modeling). Unless otherwise specified, we measured steps to target on the validation set. We chose the target metric values based on an initial set of experiments with practical computational budgets. For each model, task, optimizer, and batch size, we independently tuned the learning rate $\alpha$, the parameters governing the learning rate schedule (where applicable), and optimizer-specific metaparameters (see Appendix F.4). We manually chose the search spaces based on our initial experiments, and we verified after each experiment that the optimal metaparameter values were far from the search space boundaries. We used quasi-random search [Bousquet et al., 2017] to tune the metaparameters with fixed budgets of non-divergent[4] trials (100 for Simple CNN, ResNet8, and Transformer, and 200 for ResNet32 and VGG11). We chose the trial that reached the target metric value using the fewest number of steps.

### 4.1 Critical Batch Size Depends on the Optimizer

Figure 5 shows the relationship between batch size and steps to target for each model, task, and optimizer. In each case, as the batch size grows, there is an initial period of perfect scaling where doubling the batch size halves the steps to target, but once the batch size exceeds a problem-dependent critical batch size, there are rapidly diminishing returns, matching the results of [Goyal et al., 2017, McCandlish et al., 2018, Shallue et al., 2018]. K-FAC has the largest critical batch size in all cases, highlighting the usefulness of preconditioning. Momentum SGD extends perfect scaling to larger batch sizes than plain SGD, but for batch sizes smaller than the plain SGD critical batch size, momentum SGD requires as many steps as plain SGD to reach the target. This is consistent with both the empirical results of Shallue et al. [2018] and our NQM simulations. By contrast, Adam and K-FAC can reduce the number of steps needed to reach the target compared to plain SGD even for the smallest batch sizes, although neither optimizer does so in all cases. Finally, we see some evidence that the benefit of momentum diminishes with preconditioning (Figures 5a and 5b), as predicted by our NQM simulations, although we do not see this in all cases (e.g. Figure 5c and 5f).

## 4.2 Exponential Moving Average Improves Convergence with Minimal Computation Cost

To verify the predictions of NQM on exponential moving average (EMA), we conducted some experiments on comparing EMA with plain SGD. We follow the same protocol of Figure 5 and report the results in Figure 6. As expected, the results on real neural networks closely match our predictions based on NQM analysis. In particular, SGD with EMA appears to reach the

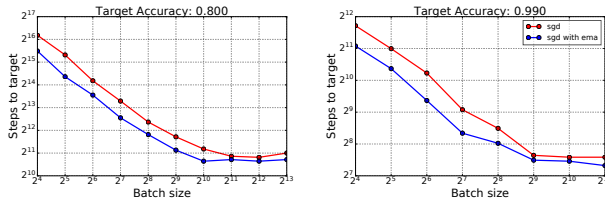

**Figure 6:** Steps to training accuracy versus batch size. **Left:** ResNet8 on CIFAR10; **Right**: Simple CNN on MNIST.

same target with fewer steps than plain SGD at small batch sizes, though the benefit of EMA diminishes with large batch sizes. Besides, we note that EMA leads to smaller critical batch sizes and achieves the same acceleration with less computation.

## 4.3 Optimal Learning Rate

The NQM predicts that the optimal constant learning rate for plain SGD (or effective learning rate for momentum SGD) scales linearly with batch size initially, and then levels off after a certain batch size. Figure 7 shows the empirical optimal (effective) learning rate as a function of batch size for simple CNN on MNIST and ResNet8

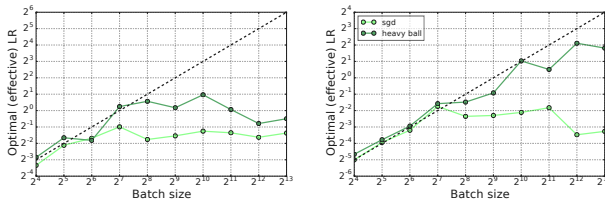

**Figure 7:** Optimal learning rates for plain SGD and momentum SGD. **Left**: Simple CNN on MNIST; **Right**: ResNet8 on CIFAR10

on CIFAR10. For small batch sizes, the optimal learning rate of plain SGD appears to match the optimal effective learning rate of momentum SGD. However, after a certain batch size, the optimal learning rate for plain SGD saturates while the optimal effective learning rate of momentum SGD keeps increasing. Interestingly, plain SGD and momentum SGD appear to deviate at the same batch size in the optimal effective learning rate and steps to target plots (Figures 5 and 7).

## 4.4 Steps to Target on the Training Set

Figure 8 shows the empirical relationship between batch size and steps to target, measured on the training set, for ResNet8 and ResNet32 on CIFAR10. For ResNet8, the curves are almost identical to those using validation accuracy (Figure 5c), but for ResNet32, the gaps between different optimizers become much smaller than in Figure 5e and the effects of momen-

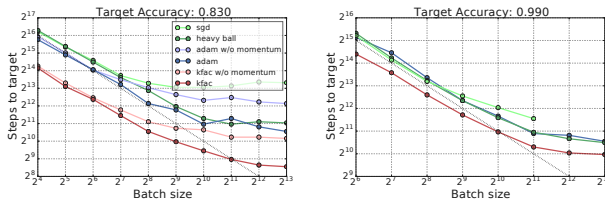

**Figure 8:** Steps to training accuracy versus batch size on CIFAR10. **Left:** ResNet8; **Right**: ResNet32.

tum and preconditioning appear to become less significant. Nevertheless, the qualitative differences between optimizers are consistent with the validation set measurements.

## 5 Conclusion

In this work, we analyzed the interactions between the batch size and the optimization algorithm from two perspectives: experiments with real neural networks, and a noisy quadratic model with parameters chosen based on empirical observations about neural networks. Despite its simplicity, the noisy quadratic model agrees remarkably well with a variety of neural network training phenomena, including learning rate scaling, critical batch sizes, and the effects of momentum, preconditioning and averaging. More importantly, the noisy quadratic model allows us to run experiments in seconds, while it can take weeks, or even months, to conduct careful large-scale experiments with real neural networks. Therefore, the noisy quadratic model is a convenient and powerful way to quickly formulate testable predictions about neural network optimization.

## Acknowledgements

RG acknowledges support from the CIFAR Canadian AI Chairs program and the Ontario MRIS Early Researcher Award.

## Footnotes

[2] To see this, note that the term in the square root of eqn. (7) for $r_1$ can be written as $(1 - \beta - (1+\beta)\alpha h_i/1-\beta)^2 + \mathcal{O}(\alpha^2 h_i^2)$. Dropping the $\mathcal{O}(\alpha^2 h_i^2)$ term and simplifying gives the claimed expression for $r_1$.

[3]For a given schedule and number of time steps, we obtain the exact risk using dynamic programming with eqn. (3). For stability, the learning rates are constrained to be at most $2/h_1$. For a fixed number of time steps, we minimize this risk using BFGS. We determine the optimal number of time steps using binary search.

[4]We discarded trials with a divergent training loss, which occurred when the learning rate was too high.

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
