[Supplementary Material · NQM_supplement.pdf]

# A  Kronecker-factored Approximate Curvature (K-FAC)

Kronecker-factored approximate curvature (K-FAC) [Martens and Grosse, 2015] uses a Kronecker-factored approximation to the curvature matrix to perform efficient approximate natural gradient updates. Considering the $l$-th layer in a neural network whose input activations are $\mathbf{a} \in \mathbb{R}^n$, weight matrix $\mathbf{W} \in \mathbb{R}^{n \times m}$, and outputs $\mathbf{s} \in \mathbb{R}^m$, we have $\mathbf{s} = \mathbf{W}^\top \mathbf{a}$. Therefore, the weight gradient is $\nabla_{\mathbf{W}} \mathcal{L} = \mathbf{a}(\nabla_{\mathbf{s}} \mathcal{L})^\top$. With this formula, K-FAC decouples this layer's Fisher matrix $\mathbf{F}$ using an independence assumption:

$$
\begin{aligned}
\mathbf{F} &= \mathbb{E}[\mathrm{vec}\{\nabla_{\mathbf{W}} \mathcal{L}\}\mathrm{vec}\{\nabla_{\mathbf{W}} \mathcal{L}\}^\top] = \mathbb{E}[\{\nabla_{\mathbf{s}} \mathcal{L}\}\{\nabla_{\mathbf{s}} \mathcal{L}\}^\top \otimes \mathbf{a}\mathbf{a}^\top] \\
&\approx \mathbb{E}[\{\nabla_{\mathbf{s}} \mathcal{L}\}\{\nabla_{\mathbf{s}} \mathcal{L}\}^\top] \otimes \mathbb{E}[\mathbf{a}\mathbf{a}^\top] = \mathbf{S} \otimes \mathbf{A}
\end{aligned}
\tag{13}
$$

where $\mathbf{A} = \mathbb{E}[\mathbf{a}\mathbf{a}^\top]$ and $\mathbf{S} = \mathbb{E}[\{\nabla_{\mathbf{s}} \mathcal{L}\}\{\nabla_{\mathbf{s}} \mathcal{L}\}^\top]$. Decomposing $\mathbf{F}$ into $\mathbf{A}$ and $\mathbf{S}$ not only avoids the quadratic storage cost of the exact Fisher, but also enables tractable computation of the approximate natural gradient:

$$
\begin{aligned}
\mathbf{F}^{-1}\mathrm{vec}\{\nabla_{\mathbf{W}} \mathcal{L}\} &= (\mathbf{S}^{-1} \otimes \mathbf{A}^{-1})\,\mathrm{vec}\{\nabla_{\mathbf{W}} \mathcal{L}\} \\
&= \mathrm{vec}[\mathbf{A}^{-1}\nabla_{\mathbf{W}} \mathcal{L} \mathbf{S}^{-1}]
\end{aligned}
\tag{14}
$$

As shown by eqn. (14), computing natural gradient using K-FAC only consists of matrix transformations comparable to size of $\mathbf{W}$, making it very efficient.

Later, Grosse and Martens [2016] further extended K-FAC to convolutional layers under additional assumptions of spatial homogeneity (**SH**) and spatially uncorrelated derivatives (**SUD**). Suppose the input $\mathbf{a} \in \mathbb{R}^{c_{\mathrm{in}} \times h \times w}$ and the output $\mathbf{s} \in \mathbb{R}^{c_{\mathrm{out}} \times h \times w}$, then the gradient of the reshaped weight $\mathbf{W} \in \mathbb{R}^{c_{\mathrm{out}} \times c_{\mathrm{in}} k^2}$ is $\nabla_{\mathbf{W}} \mathcal{L} = \sum \mathbf{a}_i \nabla_{\mathbf{s}_i} \mathcal{L}^\top$, and the corresponding Fisher matrix is:

$$
\begin{aligned}
\mathbf{F} &\approx \sum \mathbb{E}\left[\{\nabla_{\mathbf{s}_i} \mathcal{L}\}\{\nabla_{\mathbf{s}_{i'}} \mathcal{L}\}^\top\right] \otimes \mathbb{E}\left[\mathbf{a}_i \mathbf{a}_{i'}^\top\right] \\
&\approx \underbrace{\left(\frac{1}{|\mathcal{I}|}\sum \mathbb{E}\left[\{\nabla_{\mathbf{s}_i} \mathcal{L}\}\{\nabla_{\mathbf{s}_i} \mathcal{L}\}^\top\right]\right)}_{\mathbf{S},\,\mathrm{size}=(c_{\mathrm{out}})^2} \otimes \underbrace{\left(\sum \mathbb{E}\left[\mathbf{a}_i \mathbf{a}_i^\top\right]\right)}_{\mathbf{A},\,\mathrm{size}=(c_{\mathrm{in}} \times k^2)^2}
\end{aligned}
\tag{15}
$$

where $\mathcal{I} = [h] \times [w]$ is the set of spatial locations, $\mathbf{a}_i \in \mathbb{R}^{c_{\mathrm{in}} k^2}$ is the patch extracted from $\mathbf{a}$, $\nabla_{\mathbf{s}_i} \mathcal{L} \in \mathbb{R}^{c_{\mathrm{out}}}$ is the gradient to each spatial location in $\mathbf{s}$ and $i, i' \in \mathcal{I}$.

## A.1  K-FAC for Transformer

K-FAC has been implemented on the autoencoder [Martens and Grosse, 2015] and various convolutional networks [Grosse and Martens, 2016, Ba et al., 2017] before. To our knowledge, this is the first time K-FAC is implemented on the Transformer model. What is different from the previous models is the shared weight matrix between the embedding layer and the pre-softmax linear transformation [Vaswani et al., 2017]. In particular, the weight matrix is transposed at the pre-softmax layer: $\mathbf{s} = \mathbf{W}\mathbf{a}$ and $\nabla_{\mathbf{W}} \mathcal{L} = (\nabla_{\mathbf{s}} \mathcal{L})\mathbf{a}^\top$. With the same assumptions as the non-transposed case, we get

$$
\mathbf{F} \approx \mathbb{E}[\mathbf{a}\mathbf{a}^\top \otimes \{\nabla_{\mathbf{s}} \mathcal{L}\}\{\nabla_{\mathbf{s}} \mathcal{L}\}^\top] = \mathbf{A} \otimes \mathbf{S}
\tag{16}
$$

i.e. the positions of the two Kronecker factors are swapped. If we name the two Kronecker factors "input factor" and "output factor" respectively, i.e. $\mathbf{F} \approx input\_factor \otimes output\_factor$, then for the weight matrix that is shared between the embedding layer and the pre-softmax layer, the *input_factor* has contributions from both the embedding inputs and the gradients of pre-softmax layer outputs; and the *output_factor* has contributions from both the pre-softmax layer inputs and the gradients of the embedding outputs. In practice, when computing a Kronecker factor, we treat contribution from multiple sources as an equivalent situation as contribution from multiple training examples from a mini-batch. Also note that because of the high dimensionality of the embedding weight matrix (with a vocabulary size of 32,768), the dense input factor would have size [32768, 32768]. In order to save memory, we use a diagonal matrix to estimate the *input_factor*. The *output_factor* is still estimated with a dense matrix.

# B   Dynamics of momentum SGD on noisy quadratic model

Similar to plain SGD, by treating $\theta_i$ as a random variable, we can explicitly write down the dynamics of its expectation and variance. But due to the use of momentum, we need to take into account $m_i$ and its correlation with $\theta_i$. Because each dimension evolves independently, we drop the the dimension subscripts. We first calculate the expectation of the parameter and velocity:

$$\begin{aligned}
\mathbb{E}\left[\theta(t+1)\right] &= (1-\alpha h)\mathbb{E}\left[\theta(t)\right] - \alpha\beta\mathbb{E}\left[m(t)\right] \\
\mathbb{E}\left[m(t+1)\right] &= \beta\mathbb{E}\left[m(t)\right] + h\mathbb{E}\left[\theta(t)\right]
\end{aligned} \tag{17}$$

We then calculate the variance:

$$\begin{aligned}
\mathbb{V}\left[\theta(t+1)\right] &= (1-\alpha h)^2\mathbb{V}\left[\theta(t)\right] + (\alpha\beta)^2\mathbb{E}\left[m(t)\right] - 2(1-\alpha h)\alpha\beta\mathrm{Cov}(t) + \frac{\alpha^2 c}{B} \\
\mathbb{V}\left[m(t+1)\right] &= \beta^2\mathbb{V}\left[m(t)\right] + h^2\mathbb{V}\left[\theta(t)\right] + 2\beta h\mathrm{Cov}(t) + \frac{c}{B}
\end{aligned} \tag{18}$$

where $\mathrm{Cov}(t) = \mathrm{Cov}(\theta(t), m(t))$ evolves as

$$\mathrm{Cov}(t+1) = (1-\alpha h)h\mathbb{V}\left[\theta(t)\right] - \alpha\beta^2\mathbb{V}\left[m(t)\right] + (1-2\alpha h)\beta\mathrm{Cov}(t) - \frac{\alpha c}{B} \tag{19}$$

Because the expected risk is totally decided by $\mathbb{E}\left[\theta\right]^2 + \mathbb{V}\left[\theta\right]$, we define $A(\cdot) = \mathbb{E}\left[\cdot\right]^2 + \mathbb{V}\left[\cdot\right]$ and $C(t) = \mathbb{E}[\theta(t)]\mathbb{E}[m(t)] + \mathrm{Cov}(\theta(t), m(t))$. We can then simplify the dynamics as follows

$$\begin{aligned}
A(\theta(t+1)) &= (1-\alpha h)^2 A(\theta(t)) + (\alpha\beta)^2 A(m(t)) - 2(1-\alpha h)\alpha\beta C(t) + \frac{\alpha^2 c}{B} \\
A(m(t+1)) &= \beta^2 A(m(t)) + h^2 A(\theta(t)) + 2\beta h C(t) + \frac{c}{B} \\
C(t+1) &= (1-\alpha h)h A(\theta(t)) - \alpha\beta^2 A(m(t)) + (1-2\alpha h)\beta C(t) - \frac{\alpha c}{B}
\end{aligned} \tag{20}$$

or equivalently

$$\underbrace{\begin{bmatrix} A(\theta(t+1)) \\ \alpha^2 A(m(t+1)) \\ -\alpha C(t+1) \end{bmatrix}}_{\mathbf{v}(t+1)} = \underbrace{\begin{bmatrix} (1-\alpha h)^2 & \beta^2 & 2(1-\alpha h)\beta \\ (\alpha h)^2 & \beta^2 & -2\beta\alpha h \\ -(1-\alpha h)\alpha h & \beta^2 & (1-2\alpha h)\beta \end{bmatrix}}_{\text{transition matrix } \mathbf{T}} \underbrace{\begin{bmatrix} A(\theta(t)) \\ \alpha^2 A(m(t)) \\ -\alpha C(t) \end{bmatrix}}_{\mathbf{v}(t)} + \underbrace{\begin{bmatrix} \frac{\alpha^2 c}{B} \\ \frac{\alpha^2 c}{B} \\ \frac{\alpha^2 c}{B} \end{bmatrix}}_{\mathbf{n}} \tag{21}$$

The convergence rate is determined by the transition matrix $\mathbf{T}$ which has the characteristic polynomial

$$|\mathbf{T} - \lambda\mathbf{I}| = -(\lambda - \beta)(\lambda^2 - (\beta^2 - 2\alpha h\beta + (1-\alpha h)^2)\lambda + \beta^2) \tag{22}$$

With the momentum value $\beta = (1-\sqrt{\alpha h})^2$, all eigenvalues of the transition matrix are equal to each other with the value $\beta$, giving the fastest convergence.

## C   Proof of Theorem 1

For a linear dynamical system like eqn. (21), we can get $\mathbf{v}(t)$ in the following form:

$$\mathbf{v}(t) = \mathbf{T}^t\mathbf{v}(0) + \sum_{p=1}^{t+1}\mathbf{T}^{p-1}\mathbf{n} \le \mathbf{T}^t\mathbf{v}(0) + \sum_{p=1}^{\infty}\mathbf{T}^{p-1}\mathbf{n} \tag{23}$$

We first analyze the stochastic term $\sum_{p=1}^{\infty}\mathbf{T}^{p-1}\mathbf{n}$. For notational convenience, we define

$$\sum_{p=1}^{\infty}\mathbf{T}^{p-1}\mathbf{n} \triangleq \sum_{p=0}^{\infty}[x_p, y_p, z_p]^\top \tag{24}$$

In eqn. (24), we append zero vector $[x_0, y_0, z_0]^\top$ for convenience. To compute the infinite sum, we first focus on a single term. We have the following update:

$$\begin{aligned}
\sqrt{x_{p+1}} &= (1-\alpha h)\sqrt{x_p} + \beta\sqrt{y_p} \\
\sqrt{y_{p+1}} &= -\alpha h\sqrt{x_p} + \beta\sqrt{y_p}
\end{aligned} \tag{25}$$

Since we only care $x_p$ which totally decide the loss, so we get rid of $y_p$ by merging two updates, which yields a second-order difference equation:

$$\sqrt{x_{p+1}} = (1 - \alpha h + \beta)\sqrt{x_p} - \beta\sqrt{x_{p-1}} \tag{26}$$

with initial conditions $\sqrt{x_0} = 0$ and $\sqrt{x_1} = \sqrt{\frac{\alpha^2 c}{B}}$. To solve the second-order difference equation, we leverage the Z-transform to get the analytical form. Based on basic manipulation of the Z-transform, we have the Z-domain function

$$X(Z) = \frac{\sqrt{\frac{\alpha^2 c}{B}} Z}{Z^2 - (1 - \alpha h + \beta)Z + \beta} = \frac{\sqrt{\frac{\alpha^2 c}{B}}}{r_1 - r_2}\left(\frac{1}{1 - Z^{-1}r_1} - \frac{1}{1 - Z^{-1}r_2}\right) \tag{27}$$

where $r_1$ and $r_2$ are two roots of equation $z^2 - (1 - \alpha h + \beta)z + \beta$. Then, we use the inverse Z-transform to get $\sqrt{x_p}$:

$$\sqrt{x_p} = \sqrt{\frac{\alpha^2 c}{B}} \frac{r_1^p - r_2^p}{r_1 - r_2} \tag{28}$$

and therefore

$$x_p = \frac{\alpha^2 c}{B} \frac{r_1^{2p} + r_2^{2p} - 2(r_1 r_2)^p}{(r_1 - r_2)^2} \tag{29}$$

Now, we are ready to compute the infinite sum $\sum_{p=0}^{\infty} x_p$:

$$\begin{aligned}
\sum_{p=0}^{\infty} x_p &= \frac{\frac{\alpha^2 c}{B}}{(r_1 - r_2)^2}\left(\frac{1}{1 - r_1^2} + \frac{1}{1 - r_2^2} - \frac{2}{1 - r_1 r_2}\right) \\
&= \frac{\alpha^2 c}{B}\frac{1 + r_1 r_2}{(1 - r_1^2)(1 - r_2^2)(1 - r_1 r_2)}
\end{aligned} \tag{30}$$

Because $r_1$ and $r_2$ are two roots with $r_1 r_2 = \beta$, $r_1 + r_2 = 1 - \alpha h + \beta$, we have

$$\sum_{p=0}^{\infty} x_p = \frac{\alpha c(1 + \beta)}{Bh(2\beta + 2 - \alpha h)(1 - \beta)} \tag{31}$$

Now, we analyze the deterministic term. Similar to the analysis of stochastic term, we have the same second-order difference equation

$$\sqrt{x'_{p+1}} = (1 - \alpha h + \beta)\sqrt{x'_p} - \beta\sqrt{x'_{p-1}} \tag{32}$$

except the initial conditions become $\sqrt{x'_0} = \sqrt{x'_1} = \sqrt{A(\theta(0))}$. According to Z-transform, we have

$$x'_t = \left(\frac{r_1^{t+1} - r_2^{t+1} - \beta(r_1^t - r_2^t)}{r_1 - r_2}\right)^2 A(\theta(0)) \tag{33}$$

Along with eqn. (31), we have

$$A(\theta(t)) \le \left(\frac{r_1^{t+1} - r_2^{t+1} - \beta(r_1^t - r_2^t)}{r_1 - r_2}\right)^2 A(\theta(0)) + \frac{\alpha c(1 + \beta)}{Bh(2\beta + 2 - \alpha h)(1 - \beta)} \tag{34}$$

# D  Proof of Theorem 2

Similar to plain SGD, by treating $\theta_i$ as a random variable, we can explicitly write down the dynamics of its expectation and variance. But due to the use of moving averaging, we need to take into account $\tilde{\theta}_i$ and its correlation with $\theta_i$. Because each dimension evolves independently, we drop the the dimension subscripts. We first calculate the expectation of the parameter and the average:

$$\begin{aligned}
\mathbb{E}\left[\theta(t+1)\right] &= (1 - \alpha h)\mathbb{E}\left[\theta(t)\right] \\
\mathbb{E}[\tilde{\theta}(t+1)] &= \gamma \mathbb{E}[\tilde{\theta}(t)] + (1 - \gamma)(1 - \alpha h)\mathbb{E}\left[\theta(t)\right]
\end{aligned} \tag{35}$$

We then calculate the variance:

$$\mathbb{V}\left[\theta(t+1)\right] = (1-\alpha h)^2 \mathbb{V}\left[\theta(t)\right] + \frac{\alpha^2 c}{B}$$

$$\mathbb{V}[\tilde{\theta}(t+1)] = \gamma^2 \mathbb{V}[\tilde{\theta}(t)] + (1-\gamma)^2(1-\alpha h)^2 \mathbb{V}\left[\theta(t)\right] \qquad (36)$$

$$+ 2\gamma(1-\gamma)(1-\alpha h)\mathrm{Cov}(t) + \frac{(1-\gamma)^2\alpha^2 c}{B}$$

where $\mathrm{Cov}(t) = \mathrm{Cov}(\theta(t),\tilde{\theta}(t))$ evolves as

$$\mathrm{Cov}(t+1) = (1-\gamma)(1-\alpha h)^2 \mathbb{V}\left[\theta(t)\right] + (1-\alpha\gamma)\mathrm{Cov}(t) + \frac{(1-\gamma)\alpha^2 c}{B} \qquad (37)$$

Because the expected risk is totally decided by $\mathbb{E}[\tilde{\theta}]^2 + \mathbb{V}[\tilde{\theta}]$, we define $A(\cdot) = \mathbb{E}\left[\cdot\right]^2 + \mathbb{V}\left[\cdot\right]$ and $C(t) = \mathbb{E}[\theta(t)]\mathbb{E}[\tilde{\theta}(t)] + \mathrm{Cov}(\theta(t),\tilde{\theta}(t))$. We can then simplify the dynamics as follows

$$\underbrace{\begin{bmatrix} A(\theta(t+1)) \\ \frac{A(\tilde{\theta}(t+1))}{(1-\gamma)^2} \\ \frac{C(t+1)}{(1-\gamma)} \end{bmatrix}}_{\mathbf{v}(t+1)} = \underbrace{\begin{bmatrix} (1-\alpha h)^2 & 0 & 0 \\ (1-\alpha h)^2 & \gamma^2 & 2\gamma(1-\alpha h) \\ (1-\alpha h)^2 & 0 & \gamma(1-\alpha h) \end{bmatrix}}_{\text{transition matrix } \mathbf{T}} \underbrace{\begin{bmatrix} A(\theta(t)) \\ \frac{A(\tilde{\theta}(t))}{(1-\gamma)^2} \\ \frac{C(t)}{(1-\gamma)} \end{bmatrix}}_{\mathbf{v}(t)} + \underbrace{\begin{bmatrix} \frac{\alpha^2 c}{B} \\ \frac{\alpha^2 c}{B} \\ \frac{\alpha^2 c}{B} \end{bmatrix}}_{\mathbf{n}} \qquad (38)$$

For such a linear dynamical system, we can easily get the $\mathbf{v}(t)$ in the following form:

$$\mathbf{v}(t) = \mathbf{T}^t\mathbf{v}(0) + \sum_{p=1}^{t+1}\mathbf{T}^{p-1}\mathbf{n} \leq \mathbf{T}^t\mathbf{v}(0) + \sum_{p=1}^{\infty}\mathbf{T}^{p-1}\mathbf{n} \qquad (39)$$

Now, to get the closed-form of $\mathbf{v}(t)$, we first analyze the second term which involves the infinite sum. For notational convenience, we introduce the following notations:

$$\sum_{p=1}^{\infty}\mathbf{T}^{p-1}\mathbf{n} \triangleq \sum_{p=0}^{\infty}[x_p, y_p, z_p]^\top \qquad (40)$$

In eqn. (40), we append zero vector $[x_0, y_0, z_0]^\top$ for convenience. To compute the infinite sum, we first focus on a single term. We have the following update:

$$\sqrt{x_{p+1}} = (1-\alpha h)\sqrt{x_p}$$

$$\sqrt{y_{p+1}} = (1-\alpha h)\sqrt{x_p} + \gamma\sqrt{y_p} \qquad (41)$$

Since we only care $y_p$ which totally decide the loss, so we get rid of $x_p$ by merging two updates, which yields a second-order difference equation:

$$\sqrt{y_{p+1}} = (1-\alpha h + \gamma)\sqrt{y_p} - (1-\alpha h)\gamma\sqrt{y_{p-1}} \qquad (42)$$

with initial conditions $\sqrt{y_0} = 0$ and $\sqrt{y_1} = \sqrt{\frac{\alpha^2 c}{B}}$. To solve the second-order difference equation, we leverage the Z-transform to get the analytical form. Based on basic manipulation of the Z-transform, we have the Z-domain function

$$Y(Z) = \frac{\sqrt{\frac{\alpha^2 c}{B}}Z}{Z^2 - (1-\alpha h + \gamma)Z + \gamma} = \frac{\sqrt{\frac{\alpha^2 c}{B}}}{r_1 - r_2}\left(\frac{1}{1 - Z^{-1}r_1} - \frac{1}{1 - Z^{-1}r_2}\right) \qquad (43)$$

where $r_1$ and $r_2$ are two roots of equation $z^2 - (1-\alpha h + \gamma)z + (1-\alpha h)\gamma$. Then, we use the inverse Z-transform to get $\sqrt{y_p}$:

$$\sqrt{y_p} = \sqrt{\frac{\alpha^2 c}{B}}\frac{r_1^p - r_2^p}{r_1 - r_2} \qquad (44)$$

and therefore

$$y_p = \frac{\alpha^2 c}{B}\frac{r_1^{2p} + r_2^{2p} - 2(r_1 r_2)^p}{(r_1 - r_2)^2} \qquad (45)$$

Now, we are ready to compute the infinite sum $\sum_{p=0}^{\infty} y_p$:

$$
\begin{aligned}
\sum_{p=0}^{\infty} y_p &= \frac{\frac{\alpha^2 c}{B}}{(r_1 - r_2)^2} \left( \frac{1}{1 - r_1^2} + \frac{1}{1 - r_2^2} - \frac{2}{1 - r_1 r_2} \right) \\
&= \frac{\alpha^2 c}{B} \frac{1 + r_1 r_2}{(1 - r_1^2)(1 - r_2^2)(1 - r_1 r_2)}
\end{aligned}
\tag{46}
$$

It is easy to see that $r_1 = 1 - \alpha h$ and $r_2 = \gamma$, we then plug them back into eqn. (46) and get

$$
\sum_{p=0}^{\infty} y_p = \frac{\alpha c(1 + (1 - \alpha h)\gamma)}{Bh(2 - \alpha h)(1 - \gamma^2)(1 - (1 - \alpha h)\gamma)}
\tag{47}
$$

For the other term $\mathbf{T}^t \mathbf{v}(0)$, we can reuse the same second-order difference equation (42) except with initial conditions $\sqrt{y_0} = \sqrt{y_1} = \frac{1}{1-\gamma} \sqrt{A(\theta(0))}$. According to Z-transform, we have

$$
y_t = \frac{1}{(1-\gamma)^2} \left( \frac{(r_1^{t+1} - r_2^{t+1}) - \gamma(1 - \alpha h)(r_1^t - r_2^t)}{r_1 - r_2} \right)^2 A(\theta(0))
\tag{48}
$$

Therefore, we have the following upper bound:

$$
A(\tilde{\theta}(t)) \leq \left( \frac{r_1^{t+1} - r_2^{t+1} - \gamma(1 - \alpha h)(r_1^t - r_2^t)}{r_1 - r_2} \right)^2 A(\theta(0)) + \frac{\alpha c(1 - \gamma)(1 + (1 - \alpha h)\gamma)}{Bh(2 - \alpha h)(1 + \gamma)(1 - (1 - \alpha h)\gamma)}
\tag{49}
$$

# E   More results on the NQM

## E.1   Eigenspectra of Neural Networks

The main objective of this section is to examine the loss surface of modern neural networks in different stages of training in order to justify the assumptions made in NQM. Nevertheless, it is hard to visualize such a high dimensional space. Following recent work [Sagun et al., 2016, Ghorbani et al., 2019], we instead focus on analyzing the eigenspectrum of the Hessian/Fisher matrices. The Hessian/Fisher of the training loss (with respect to the parameters) is crucial in determining many behaviors of neural networks. The eigenvalues of the Hessian/Fisher characterize the local curvature of the loss surface which determines many training behaviors, including first-order methods optimization rates (at least for convex problems.)

**Figure 9:** Eigenspectra of the K-FAC approximate Fisher matrix of ResNet8 at different training iterations. The model is trained on CIFAR-10 with batch size 3000.

It has been noted that the *true* Fisher matrix is equivalent to the generalized Gauss-Newton Hessian matrix [Martens, 2014], so we take it as a proxy of the Hessian. To construct the eigenspectrum of the true Fisher matrix, we first leverage the Kronecker-factored approximation of the Fisher to get an estimation of the eigenspectrum, which may shed light upon the true eigenspectrum. Specifically, we train the network with K-FAC and then perform eigen-decomposition on saved Kronecker factors of the Fisher to calculate the eigenvalues.

The eigenspectra are plotted in Figure 9. One interesting observation is that there are only a few large eigenvalues and a few small eigenvalues in the approximate Fisher matrices; the bulk of eigenvalues are in the middle of the spectrum. We also note that after 200 iterations of training the eigenspectrum remains mostly unchanged.

## E.2   Gradient Covariance in the Kronecker-Factored Eigenbasis

To verify the assumption in Section 3.5 that $\mathbf{H}$ and $\mathbf{C}$ are codiagonalizable, we test it on practical neural networks by comparing the gradient variance to the curvature. This assumption is motivated by theoretical considerations that suggest $\mathbf{H} \approx \mathbf{C}$ for neural network training [Martens, 2014]. Ideally,

**(a)** ResNet8

**(b)** Transformer

**Figure 10: Scatter plots of second moment v.s. variance of gradients.** The gradients are projected onto the Kronecker-factored eigenbasis, which approximates the eigenbasis of the true Fisher. Each point compares the gradient variance and the second moment of the gradient in the direction of an eigenvector of the K-FAC approximated Fisher.

we would like to compare the gradient variance and the curvature of the Fisher in the directions of the eigenvectors of the true Fisher. However, it is typically infeasible to get all these eigenvectors, especially for low curvature directions. To resolve this we instead use the Kronecker-factored eigenbasis [George et al., 2018, Bae et al., 2018, Wang et al., 2019], which is obtained from the K-FAC approximation. For this experiment, we are not relying on this basis being an accurate approximation to the eigendecomposition of the true Fisher; rather, we use the eigenbasis only as a way to obtain a diverse set of directions with both high and low curvature. For a given eigenvector $\mathbf{v}$, we project the gradients $\mathbf{g}$ of each training example onto $\mathbf{v}$ and compute the gradient variance $\mathrm{Cov}(\mathbf{v}^\top \mathbf{g})$, as well as the curvature $\mathbf{v}^\top \mathbf{F} \mathbf{v}$. (The latter quantity can be obtained using matrix-vector products [Schraudolph, 2002].) As shown in Figure 10, the gradient variances closely match the curvature (especially for the ResNet8 model on CIFAR10), validating our assumption that $\mathbf{H} \approx \mathbf{C}$.

### E.3 Plots for the Evolution of the First Term in Eqn. (6)

**Figure 11: Comparison in convergence between momentum SGD and SGD with adjusted learning rate.** This plot shows values for the first term in eqn. (6) as a function of $(1 - \beta)$, which is the scaling between the "effective learning rate" and the true learning rate for momentum SGD. The red curves show the first term when using momentum, while the blue curves show the first term when using plain SGD with the learning rate set to the effective learning rate of momentum.

In Section 3.2, we claim that the convergence of momentum SGD for a single dimension is very close to that of plain SGD with an adjusted learning rate (note that we already verified that the steady state risk of momentum SGD matches plain SGD using effective learning rate in Figure 2). Here we verify this argument by comparing them in the NQM. The total risk consists of two terms (eqn. (6)): the

first term determines convergence, while the second term (steady state risk) stays constant throughout training. Given that the second stays unchanged, we only plot the first term of eqn. (6) in Figure 11. Note that the values are normalized in the figures. We observe that the convergence dynamics of the two update rules closely match each other. For this experiment we set $\alpha h = 0.0005$, but the results are not sensitive to this value.

## E.4    Verification of Eigenspectrum

In Section 3.7, we assume the diagonal entries of $\mathbf{H}$ are $\{\frac{1}{i}\}_{i=1}^{d}$. To justify this choice, we compare the K-FAC eigenspectra of ResNet8 to this distribution in Figure 12. The distribution of eigenvalues we chose for $\mathbf{H}$ in the NQM very closely matches the eigenspectra of the real neural network, validating the assumption that the diagonal entries of $\mathbf{H}$ are $\{\frac{1}{i}\}_{i=1}^{d}$ in Section 3.5.

**Figure 12: Comparison between K-FAC Fisher eigenspectra and the $\frac{1}{i}$ distribution used in the NQM.**

## E.5    Effect of Loss Threshold

Recall that a main objective of this work is to characterize the effects of increasing the batch size on training time, as measured in the number of steps necessary to reach a goal target error/loss. Here we experiment with different loss thresholds to study the relationship between batch size and number of training steps. To obtain the minimal training steps for a given batch size, we do grid search over constant learning rates. Figure 13 shows that increasing the batch size initially decreases the required number of training steps proportionally, but eventually there are diminishing returns, which matches the empirical findings [Golmant et al., 2018, Shallue et al., 2018]. The shape of the curves

**Figure 13:** Number of training steps required to reach a target loss as a function of batch size for different loss threshold values.

is characteristically the same for different loss thresholds, though the critical batch size seems to increase for more difficult thresholds.

## E.6    Results of Optimal Learning Rate on NQM

**(a)** Without Momentum        **(b)** Fixed Momentum 0.9        **(c)** Tuned Momentum

**Figure 14: Optimal learning rate v.s. batch size for different preconditioning powers. (a)** When momentum is not used, the learning rate increases with batch size until it is limited by the maximum stable learning rate. Larger preconditioning powers reduce the optimal learning rate for the same batch size, thus extending the batch size where the learning rate levels off. **(b, c)** Fixed (0.9) and tuned momentum values. In (b) and (c), we plot the *effective learning rate* for momentum SGD, defined as $\frac{\alpha}{1-\beta}$. The dashed lines are the same plots from (a) for easier comparison.

## E.7 Final Learning Rate of Different Batch Sizes for PWC Learning Rate Scheme

In Section 3.7.2, we study the piecewise constant learning rate scheme. The optimal scheme starts with a high learning rate which drops later in training (Figure 3c). Recall that for fixed learning rates, we observed that the optimal learning rate scaled linearly with the batch size for small batch sizes, but it is unclear whether there is a similar phenomenon for learning rate decay. In Figure 15, we plot the final learning rate as a function of batch size and show that it also scales linearly with batch size.

**Figure 15:** Final learning rate of the piecewise-constant learning rate scheme v.s. batch size.

# F   More Details for Experiments

## F.1   Data Sets

The data sets in Table 1 (MNIST, Fashion MNIST, CIFAR10, ImageNet and LM1B) are identical to those of Shallue et al. [2018] (described in their Appendix A.1). For CIFAR10 we used data augmentation (including horizontal flip and random crop), but they did not.

## F.2   Model Details

This section provides details of models in Table 1. The models are very similar (and some identical) to those used in Shallue et al. [2018] (described in their Appendix B). Any modifications from them are highlighted in this section.

**Simple CNN** consists of 2 convolutional layers with max-pooling followed by 1 fully connected hidden layer. The convolutional layers use 5×5 filters with stride length 1, "same" padding [Goodfellow et al., 2016], and ReLU activation function. Max pooling uses 2×2 windows with stride length 2. Unlike in Shallue et al. [2018], we did not use any dropout regularization (while they used dropout with probability 0.4 in the fully connected layer). We used 32 and 64 filters in the convolutional layers and 1,024 units in the fully connected layer. This corresponds to the "base" configuration in Shallue et al. [2018].

**ResNet8** [He et al., 2016] consists of 7 convolutional layers with residual connections followed by 1 fully connected hidden layer. We used the identical architecture as Shallue et al. [2018]. In particular, we did not use batch normalization. The only difference is that we used data augmentation in our experiments.

**ResNet32** [He et al., 2016] consists of 31 convolutional layers with residual connections followed by 1 fully connected hidden layer (see Section 4.2 of He et al. [2016]). We replaced batch normalization [Ioffe and Szegedy, 2015] with ghost batch normalization to keep the training objective fixed between batch sizes and to avoid possible negative effects from computing batch normalization statistics over a large number of examples [Hoffer et al., 2017]. We used a ghost batch size of 32 for all experiments. We also applied label smoothing [Szegedy et al., 2016] to regularize the model at training time, which was helpful for larger batch sizes. We set the label smoothing parameter to 0.1 in all experiments. Instead of using weight decay, we applied channel-wise weight normalization by constraining the Frobenius norm of each convolutional channel to be exactly 1, which controls the effective learning rate [Zhang et al., 2019b, van Laarhoven, 2017].

**VGG11** [Simonyan and Zisserman, 2015] consists of 8 convolutional layers followed by 1 fully connected hidden layers. as in ResNet32, we used Ghost batch normalization, label smoothing, and channel-wise weight normalization.

**Transformer** Vaswani et al. [2017] is a self-attention model. We chose the Transformer model identical to the "base" model described in Vaswani et al. [2017], except with only two hidden layers instead of six. This is identical to the "Transformer Shallow" model in Shallue et al. [2018].

### F.3 Learning Rate Schedules

This section describes two learning rate schedules mentioned in Table 1: constant schedule and linear decay schedule. Constant schedule simply keeps a fixed learning rate throughout training:

$$\alpha(t) = \alpha_0,$$

where $t$ is the training step index. Linear decay schedule is

$$\alpha(t) = \alpha_0 - (1 - \gamma)\frac{t}{T},$$

where $\alpha_0$ is the initial learning rate, $\gamma$ is the rate of decay, and $T$ is the number of steps taken to reach the final learning rate. Shallue et al. [2018] experimented with various learning rate schedules and found that linear decay matched performance of the other schedules with fewer hyperparameters to tune. Therefore, we also chose the linear decay schedule, for which we tuned $\alpha_0$, $\gamma$ and $T$.

### F.4 Optimizer-Specific Hyperparamters

For momentum SGD, we tuned the momentum $\beta$. For Adam, we tuned $\beta_1$, $\beta_2$, and $\epsilon$ (see Kingma and Ba [2014]). For K-FAC, we tuned damping and the trust region constraint (also known as the KL clipping term) for Transformer, keeping momentum $= 0.9$ and the moving average parameter for damping $= 0.99$; for all other models, we tuned all four parameters (see Martens and Grosse [2015]).