[Reviews · NeurIPS 2019]

Reviewer 1



After reading the rebuttal and the other reviews, my score stays the same. Please add the discussed clarifications to the paper. ======== Overall I liked the submission, mostly for the well thought-out experiments that highlight an interesting and useful phenomenon: that the linear scaling rule for batch size extends farther when momentum or preconditioning are used. The theory also led to intuitive, well-presented results that were predictive of the experimental behavior. This is a new, significant result for the community, and overall I recommend acceptance. There are a couple of changes that would substantially improve the paper: First, in trying to motivate their co-diagonalization assumptions, as well as their theoretical assumption that they precondition by a power of the Hessian, the authors frequently conflate the Hessian and Fisher matrices. This conflation also appears in the study of the eigenvalues of the Fisher (in the appendix but referenced in the main text), which they pass off as eigenvalues of the Hessian. As a result some conclusions are misleading as currently stated. This would be remedied by formally defining each and changing the language around the two terms to make it more clear they are different. (To be clear: I think it is perfectly fine to make the assumptions the authors make here, since they do not rely upon the results theoretically, but rather treat them as predictions of what may (and apparently does) happen in practice) Second, the authors should be more clear about the precise forms of the algorithms they study. In particular, the authors state and prove results regarding momentum without actually stating what SGD with momentum is. Yes, this is common knowledge, but still imperative to have in the paper for the reader to follow precisely what you are doing. The same goes for later on in the experiments where preconditioning and momentum are studied together -- without an explicit algorithm, it is impossible to know e.g. in which order momentum and the preconditioned are being applied.

Reviewer 2



This paper studies how the critical batch size changes based on properties of the optimization algorithm, including SGD, momentum, and preconditioning. Theoretically, the authors analyzed the effect of batch size and learning rate via a simple quadratic model. Empirically, the authors investigated deep learning experiments and confirmed their theoretical findings for the simplified model. The paper is clearly written. The theoretical developments are based on a simple diagonalized quadratic model. 1. Regarding the proposed quadratic model, as the Hessian H is diagonal, the loss can be decomposed dimension-wise and therefore the optimization in each dimension evolves independently. These formulations are very restricted and far from the practice of deep learning models. Can the author comment on the generalizability of the analysis to non-diagonal quadratic models or even simple neural network models? 2. Regarding the optimizers, the author adds some Gaussian noise to the gradients to model the effect of stochastic sampling. Such a noise model is fixed throughout optimization and has diagonal covariance matrix, which is different from SGD whose stochastic noise also evolves and the covariance matrix can be non-diagonal. Also, the author claims that Adam can be viewed as a preconditioned SGD, but the preconditioning matrix considered takes a very special form of H^p. While all these simplifications can lead to an analytical understanding of the optimization, they do not necessarily cover the practice in training deep models. 3. Overall I think the theoretical contribution of this paper is limited. The author tries to explain and understand the optimization mechanism of deep learning by studying a simplified quadratic model. The deep learning scenario violates many of the assumptions in this paper, e.g., diagonal Hessian, independent optimization among dimensions, fixed noise, etc. While the theoretical results of the quadratic model fit (to some extent) the empirical observations in training deep models, there is no good justification for a solid connection between these two models. It would be better if the authors can justify (to some extent) the motivation of simplifying deep learning scenarios into such a simple one. I have read the authors' response. It addresses my concerns on the diagonal assumption of the Hessian of NQM. Overall, I think this is an interesting paper that tries to model the relationship among the optimal choices of hyper-parameters for different optimizers in training neural networks. I am still a bit concerned about the use of a convex quadratic model and its generality. I raised my score to be 6 marginally above the threshold.

Reviewer 3



[Edit after the author feedback]: I thank the authors for addressing my comments during the author feedback. I have read the authors' response as well as the other reviews. The authors' response addresses my concerns on the simplicity of NQM. Overall,  I think this submission is interesting and provides a different direction to understand neural network optimization. I am happy to raise my rating. ========================================================== Summary: Motivated by recent various batch size phenomena in neural network training, this paper proposes a simple noisy quadratic model (NQM) to capture/predict the features of several optimization algorithms with different batch sizes in neural network training. The experimental results demonstrate the effectiveness of predictions of NQM on image classification tasks and the language modeling task. Also, the experimental results are consistent with previous studies. Pros: - The proposed model in this paper is well aligned with various phenomena in deep neural network training, which could be an interesting direction to study the role of batch size in training deep neural nets. - Empirically, this paper provides detail and importance analysis on the connection between NQM and different optimization algorithms. - The experiments are extensive, clear, and well designed, which characterize the key features of the batch size effect. Limitation: - Although the proposed NQM agrees well with previous empirical results (i.e., Goyal et al. [2017], Shallue et al. [2018]), as the objective function in (1) is convex and the quadratic form is diagonal, I think the model is not powerful enough to explain those phenomena in neural network training. Questions: - As shown in Eq (1), the objective function is very simple. Could the authors provide more explanations to justify the problem setup? As the problem is much easier than training deep neural networks in real applications. - L95, 'we assume without loss of generality that the quadratic form is diagonal'. Could you explain more on the diagonal quadratic form? There is an omission in the related work on large batch training: https://arxiv.org/abs/1709.05011 https://arxiv.org/abs/1904.00962

[Author Response · NeurIPS 2019]

We thank all the reviewers for the helpful comments and questions. Before moving on to answer reviewers' questions, we first clarify the main **contributions** and **motivations** of our paper.

The goal of the paper is to better understand neural network optimization, especially the interactions between algorithmic choices and batch-size. We proposed a simple, yet useful toy model – Noisy Quadratic Models. This model captures the essential behavior we see in real neural networks (as validated by our experiments) while **allowing us to run experiments in seconds**. Note that most assumptions we made are based on existing theoretical results or experiments in real neural networks. We stress that NQM makes it easy to test new empirical ideas for practitioners and derive new, testable theoretical results for theorists (as an example, we studied the role of momentum analytically, showing that the momentum and learning rate are interchangeable in the small batch regime, which is non-trivial in stochastic setting).

We now address all comments and questions in order.

**Interaction of momentum and preconditioning (R1).** For neural net experiments, we applied preconditioning before the momentum. For the NQM, with fixed preconditioning, both methods are equivalent. We will update the paper to clarify these points.

**Relationship between the Hessian and Fisher matrices (R1).** For the models we study, the Fisher matrix is equivalent to the Gauss-Newton Hessian; see Martens, (2014) for details. We will make this explicit in the paper.

**Assumption of diagonal Hessian (R2, R3).** The assumption that the Hessian is diagonal can be made *without loss of generality* in the sense that (as noted in lines 94–98) we focus on algorithms such as SGD and Heavy-ball momentum that are invariant under variable rotations. For such algorithms, we can analyze the evolution of iterates in a basis that makes the Hessian diagonal, without changing the dynamics of the system.

**Theoretical contribution (R2).** We stress here that the analysis of momentum in stochastic setting is non-trivial. In particular, stochasticity (gradient noise) introduces extra difficulty in analyzing heavy-ball momentum. For the NQM, we showed theoretically that momentum SGD performs similarly to plain SGD in the regime of small batch sizes but helps in the large-batch regime, which also matches previous studies and our large-scale experiments.

**NQM is too simple, and some assumptions are false for neural nets (R2, R3).** When constructing a model for some phenomenon, the model does not need to be completely faithful in all respects; rather, it is the act of abstracting away inessential features that allows one to tractably analyze phenomena. Simplicity is a virtue, as long as the model captures the effects relevant to the phenomenon. One of our main contributions is showing that (empirically) features such as non-convexity, non-stationary gradient noise, etc., are not needed to explain certain neural net training phenomena which have recently received a lot of attention. Abstracting away these features will make it much easier for others to build on our work by further analyzing these phenomena. (Of course, for explaining other phenomena we don't consider, such as local optima, one might need to re-introduce features such as non-convexity.)

**Assumptions of fixed gradient noise and particular form of preconditioner (R2).** (Note that we *don't* require the noise distribution to be Gaussian.) Both assumptions are required for analytic analysis and fast simulation, yet reflect (to some extent) the reality of neural networks. Particularly, we showed in the appendix that the Fisher matrix stabilizes after a few epochs of training on ResNet, supporting the assumption of fixed gradient noise. In other words, we expect the assumption fixed gradient noise roughly captures the essence of real neural networks in throughout all of training except the initial phase.

**Quadratic loss function (R2, R3).** We stress here that there are several ways one might try to justify the use of convex quadratic models of the objective (also see lines 83-89 of our submission). First is the well-known fact that any smooth function will resemble a convex quadratic in a small enough neighborhood around a local minimizer. And recent theoretical work (Jacot et al., 2018, Du et al., 2018, Lee et al., 2019) has argued that very wide neural networks will deviate only a small distance from the initial random point in parameter space throughout training, and that they thus behave similarly to linearized networks for the purposes of training. This implies that the objective will appear "locally convex" throughout training, and that for the squared error loss will even resemble a convex quadratic. Adding further support for the "networks as linearized models" approximation is the fact that the Generalized Gauss-Newton matrix (Martens, 2014), which is the matrix of choice behind the most powerful 2nd-order neural network optimization methods, can be seen as the Hessian of the training objective under said approximation. Finally, we note that quadratic approximations also motivated Laplace approximation (MacKay, 1992) and variational inference (Zhang et al., 2018) in Bayesian neural networks.

We thank all reviewers again. We hope that our responses address your comments and concerns.

[Meta-Review · NeurIPS 2019]

The paper analysis momentum SGD on the noisy quadratic model and provides empirical results looking at varying batch size, momentum, ane preconditioning. Well written paper. Also, reviewers had several suggestions but saw the insight that the idealized model provides to be useful.